

# Prevalence and risk factors of *Helicobacter pylori* infection among children in Kuichong Subdistrict of Shenzhen City, China

Jingjing Hu[1,*], Xiangyu Wang[1,2,*], Eng Guan Chua[3], Yongsheng He[2], Qing Shu[1], Li Zeng[1], Shiyang Luo[2], Barry J. Marshall[2,3], Aijun Liu[2] and Chin Yen Tay[2,3]

[1] The First Affiliated Hospital of Shenzhen University, Shenzhen Second People's Hospital, Shenzhen, China
[2] Shenzhen Kuichong People's Hospital, Shenzhen, China
[3] The Marshall Centre for Infectious Diseases Research and Training, University of Western Australia, Perth, WA, Australia
* These authors contributed equally to this work.

Corresponding authors
Aijun Liu, liuaijunsz@163.com
Chin Yen Tay,
alfred.tay@uwa.edu.au

## ABSTRACT

**Background:** *Helicobacter pylori* infection is a significant burden to the public health in China as it can lead to various gastric diseases including peptic ulcers and gastric cancer. Since most infections occurred during childhood, it is therefore necessary to understand the prevalence and risk determinants of this bacterial infection in children. Herewith, we conducted a cross-sectional study in the Kuichong Subdistrict of Shenzhen City to assess the prevalence and risk factors of *H. pylori* infection among children.

**Methods:** From September 2018 to October 2018, 1,355 children aged 6–12 years from four primary schools in the Kuichong Subdistrict of Shenzhen City were recruited. These children were screened for *H. pylori* infection using the $^{13}$C-urea breath test. In addition, parents were requested to fill out a standardized questionnaire. The chi-square test and multivariable logistic regression analysis were used to identify risk factors for *H. pylori*.

**Results:** Among 1,355 children recruited in this study, 226 (16.7%; 95% CI [14.7–18.7]) were positive of *H. pylori* infection. Multivariable logistic regression analysis identified six factors significantly associated with *H. pylori* infection children including parent(s) with tertiary education level (OR: 0.64; 95% CI [0.46–0.89]), testing bottle feed temperature using the mouth (OR: 1.79; 95% CI [1.19–2.68]), sharing of cutlery between the feeding person and young children during meals (OR: 1.84; 95% CI [1.22–2.78]), eating fruit after peeling (OR: 2.56; 95% CI [1.4–4.71]), frequent dining out (OR: 3.13; 95% CI [1.46–6.68]) and snacking (OR: 1.43; 95% CI [1.01–2.01]).

**Conclusions:** Overall, better educated parent(s) played a protective role against the acquisition of *H. pylori* infection in children. Testing bottle feed temperature using the mouth, cutlery sharing between the feeding person and young children, and snacking posed a lower but significant risk for *H. pylori* infection. Only eating peeled fruits and frequent dining out were associated with greater infection risks.

# INTRODUCTION

*Helicobacter pylori* is a key gastric pathogen infecting nearly half of the world's population, causing several gastric disorders including chronic gastritis, gastric atrophy, peptic and duodenal ulcerations, gastric mucosa-associated lymphoid tissue lymphoma and gastric adenocarcinoma (*Marshall & Windsor, 2005*). In China, in tandem with a high overall *H. pylori* prevalence rate of 55.8%, gastric cancer has been reported as the third most predominant cancer type and the second leading cause of cancer-related death in this nation, with an estimated 456,124 new cases and 390,182 deaths in 2018, respectively (*Feng et al., 2019*; *Hooi et al., 2017*). Based on a recent systematic review analyzing the medical expenses of Chinese patients with gastric cancer during 2002–2011, it was estimated that in 2012 the first course treatments alone would cost China nearly $3 billion USD, signifying the clinical and economic impact of *H. pylori*-related gastric diseases on the public health system (*Sun et al., 2018*).

Poor socioeconomic status, overcrowding and improper hygiene standards have been reported as risk factors for acquisition of *H. pylori* infection (*Malaty & Graham, 1994*; *Yucel, Sayan & Yildiz, 2009*). It is also believed that *H. pylori* infections mostly occur during early childhood in an intrafamilial transmission setting through both the fecal–oral and oral–oral routes as live bacterium had been previously successfully isolated from human fecal and oral sample (*Awuku et al., 2017*; *Ertem, 2013*; *Malaty et al., 2001*; *Parsonnet, Shmuely & Haggerty, 1999*; *Thomas et al., 1992*; *Urita et al., 2013*; *Yucel, Sayan & Yildiz, 2009*). Therefore, the best way of reducing *H. pylori* prevalence, perhaps, is to improve personal hygiene awareness among the primary caregivers of children, especially mothers, to prevent childhood infection.

At present and overall, there is a lack of epidemiological investigation assessing the prevalence of *H. pylori* infection among the asymptomatic children in China. Therefore, this cross-sectional study aimed to examine *H. pylori* prevalence among healthy children from Kuichong Subdistrict situated in Shenzhen, China by using the $^{13}$C-urea breath test (UBT). The $^{13}$C-UBT was selected due to its non-invasive nature and has been proven to be highly accurate in determining *H. pylori* infection status in children aged 6–12 years (*Elitsur et al., 2009*). In addition, to allow more effective strategies to be drawn for the prevention and management of *H. pylori* infection in early childhood, we also explored the relations between *H. pylori* infection rate and different socioeconomic and demographic parameters to identify the potential risk factors.

# MATERIALS AND METHODS

## Ethics approval

The study was approved by the research ethics committee of the First Affiliated Hospital of Shenzhen University, Shenzhen Second People's Hospital. Written and informed consents were obtained from all participants and their legal guardians.

## Sample size determination

According to a previous *H. pylori* prevalence study conducted in Guangzhou, we assumed that 22.9% of asymptomatic children would be infected with *H. pylori* (*Chen et al., 2007*). The samples size required for this study was estimated based on a 95% confidence interval and a 2.5% admissible error rate using the following formula:

Sample size, $n = Z_{\alpha/2}^2 \times \pi \times (1 - \pi)/\delta^2$, where $Z_{\alpha/2}$ is the 2-tailed normal deviate for an $\alpha$ level of 0.05; $\pi$ is the prevalence ratio; $\delta$ is the admission error rate.

A sample size of 1,152 was therefore required for this study. We added at least 25% to the estimate for contingency purpose, raising the final sample size to 1,355.

## Study population

To avoid selection bias, from September 2018 to October 2018, all school children aged 6–12 years from four primary schools located in Kuichong Subdistrict of Shenzhen City, China, who did not meet any of the exclusion criteria and had provided parental or guardian consent, were recruited. Exclusion criteria included intake of antibiotics within the last 4 weeks, intake of proton pump inhibitors within the last 2 weeks and previous history of *H. pylori* eradication therapy.

The [13]C-UBT (Beijing Boran Pharmaceutical, Changping, China) was performed according to the manufacturer's instructions to determine *H. pylori* infection status. All breath samples were taken and analyzed immediately in the school by the same technician. Briefly, an initial baseline breath was collected from each study participant following an overnight fast of at least 4 h. Each participant was then requested to ingest a tablet containing 75 mg [13]C labeled urea with 80–100 mL of water. After 30 min of sitting, exhalation was again collected. The [13]$CO_2$ content within the initial and 30-min expiratory air bags were analyzed using an HG-IRIS13C infrared spectrometer (Beijing Richen-Force Science & Technology Co., Beijing, China). The concentration of [13]$CO_2$ following 30 min of administration that exceeded the baseline by 4% was regarded as a positive indicator of *H. pylori* infection. The weight, height, and both head and chest circumferences of each participant were also recorded. Body mass index (BMI) was calculated as body weight divided by height squared (kg/m$^2$). The BMI category was classified according to the corresponding percentiles as recommended by The Centers for Disease Control and Prevention.

## Questionnaires

On behalf of each study participant, parents or legal guardians were requested to complete a questionnaire for data collection on sociodemographic and living conditions, birth method and childhood feeding practices, lifestyle habits and health parameters. A blank copy of the questionnaire is available as File S2. The outcomes are summarized in Table S1. Sociodemographic and living conditions consisted of age and gender of all children, number of siblings (three categories: 1, 2–3 and >3), total annual income (two categories: ≤200,000 or >200,000 RMB), household population (two categories: ≤3 or >3 persons), household space (two categories: ≤40 or >40 m$^2$/person), parents'

education level (two categories: both were high school graduates or either one had completed tertiary education) and pet ownership (two categories: yes or no).

The birth method of each child was assessed in two categories (via natural delivery or a cesarean section). Childhood feeding practices comprised the primary feeder of young children (four categories: parents, grandparents, both parents and grandparents, or other caregivers), feeding methods (three categories: feeding pre-chewed food, feeding by using shared cutleries or feeding by using a separate set of cutleries), breast feeding duration (three categories: never, up to 6 months or more than 6 months) and how the milk bottle temperature was tested (three categories: by dripping a few onto the wrist, taking a swig or using a thermometer).

Lifestyle investigations included the teeth brushing frequency (three categories: ≤3, 4–6 or ≥7 times per week), hand washing before meal and after toilet (three categories: rarely, sometimes or always), dining out frequency (three categories: ≤1, 2–4 or ≥5 times per week), how fruit was eaten (two categories: peeled or unpeeled), drinking water for consumption (two categories: raw or boiled), toothbrush sharing (two categories: yes or no), snacking regularly (two categories: yes or no) and thumb sucking (two categories: yes or no). The health conditions of each child were evaluated in three categories (≤1, 2–4 or ≥5 times per week) for bloating, abdominal discomfort, burping and diarrhea, respectively, and in two categories (present or absent) for halitosis, anemia, asthma and skin allergy, respectively.

## Statistical analysis

Unpaired, two-tailed Student's $t$-test was used for statistical comparison of continuous variables between *H. pylori* positive and negative groups. For comparison of categorical variables, the chi-square test was employed, followed by univariate logistic regression analysis using JASP software v0.11.1 (https://jasp-stats.org/). Variables significantly associated with *H. pylori* infection in the univariate test ($P < 0.05$) were selected as candidates for entry into the multivariable logistic regression models using the forward selection process. During this iterative selection process, any variable with $P > 0.1$ was eliminated. Subsequently, variables previously not included in the original model were introduced, one at a time, to identify any variable that could have an important role in *H. pylori* infection in the presence of significant variables retained earlier. The selection process was repeated but only for the additionally loaded variables until a final model was obtained. Both univariate and multivariate analyses were adjusted for age. Results were presented as odds ratio (OR) with 95% confidence intervals (CI). A $P$-value of less than 0.05 was regarded as statistically significant.

## RESULTS

### Prevalence of *H. pylori* infection

In this study, 1,355 children aged from 6 to 12 years were recruited, among which 747 (55.1%) were males and 608 (44.9%) were females (Table 1). The overall *H. pylori* prevalence rate was 16.7% (226/1355; 95% CI [14.7–18.7]). There was no distinguishable

**Table 1 Demographic characteristics and anthropometric measurements of 1,355 children recruited in this study.**

| | *H. pylori* positive | *H. pylori* negative | P |
|---|---|---|---|
| Age (years) | | | 0.032 |
| 6 (*n* = 160) | 10.6 [5.7–15.5] | 89.4 [84.5–94.3] | |
| 7 (*n* = 231) | 16.5 [11.6–21.4] | 83.5 [78.6–88.4] | |
| 8 (*n* = 207) | 23.2 [17.3–29.1] | 76.8 [70.9–82.7] | |
| 9 (*n* = 260) | 18.1 [13.3–22.9] | 81.9 [77.1–86.7] | |
| 10 (*n* = 223) | 12.6 [8.2–17] | 87.4 [83–91.8] | |
| 11 (*n* = 202) | 17.8 [12.4–23.2] | 82.2 [76.8–87.6] | |
| 12 (*n* = 72) | 16.7 [7.9–25.5] | 83.3 [74.5–92.1] | |
| Gender | | | 0.704 |
| Male (*n* = 747) | 16.3 [13.6–19] | 83.7 [81–86.4] | |
| Female (*n* = 608) | 17.1 [14–20.2] | 82.9 [79.8–86] | |
| Birthplace | | | |
| Dapeng | 15.7 [13–18.4] | 84.3 [81.6–87] | 0.265 |
| Others | 17.9 [14.8–21] | 82.1 [79–85.2] | |
| BMI | | | 0.404 |
| Underweight | 16.5 [10.2–22.8] | 83.5 [77.2–89.8] | |
| Normal | 15.8 [13.4–18.2] | 84.2 [81.8–86.6] | |
| Overweight | 18.8 [12.5–25.1] | 81.2 [74.9–87.5] | |
| Obese | 21.8 [13.6–30] | 78.2 [70–86.4] | |
| Height (cm) | 134.2 ± 11.8 | 134.5 ± 11.9 | 0.76 |
| Weight (kg) | 30.6 ± 9.6 | 30.2 ± 8.9 | 0.596 |
| Head circumference (cm) | 52.5 ± 1.7 | 52.6 ± 2.2 | 0.381 |
| Chest circumference (cm) | 65.8 ± 36.8 | 63.5 ± 20.5 | 0.369 |

Note:
Data are presented as % (95% confidence interval) or mean ± standard deviation. BMI: body mass index. The age, gender, birthplace and BMI differences between *H. pylori*-positive and negative children were compared using the chi-square test. For differences involving continuous parameters including height, weight, head circumference and chest circumference, the comparison was performed using the unpaired Student's *t*-test.

difference in *H. pylori* infection rate in gender and between children who were locally and elsewhere born. Interestingly, while 8-year-old children had the highest prevalence of *H. pylori* infection at 23.2%, lower than the overall prevalence positivity rates were seen in children aged 6 and 10 years old, at 10.6% and 12.6%, respectively. No significant differences in anthropometric measurements were observed among the infected and non-infected children.

We performed chi-square test to examine for any significant difference in different sociodemographic, lifestyle and clinical characteristics between *H. pylori*-positive and *H. pylori*-negative children. Variables tested with no significant difference between both groups are available in Table S2. As demonstrated in Table 2, the following variables contributed to greater *H. pylori* prevalence among children: having more than one child, both parents attended no education after high school, smaller living space, testing milk bottle temperature by taking a swig, feeding infant with pre-chewed food or by using

**Table 2 List of sociodemographic, lifestyle and clinical variables that had significant influence on *H. pylori* infection status in 1,355 children.**

| Variables | *H. pylori* positive | *H. pylori* negative | *p* |
|---|---|---|---|
| Number of children | | | 0.027 |
|   1 (*n* = 341) | 12.6 [9–16.2] | 87.4 [83.8–91] | |
|   2 or 3 (*n* = 918) | 17.5 [15–20] | 82.5 [80–85] | |
|   >3 (*n* = 96) | 22.9 [14.3–31.5] | 77.1 [68.5–85.7] | |
| Parents education level | | | 0.001 |
|   Both were high school graduates (*n* = 843) | 19.2 [16.5–21.9] | 80.8 [78.1–83.5] | |
|   Either one or both had completed tertiary education (*n* = 512) | 12.5 [9.6–15.4] | 87.5 [84.6–90.4] | |
| Living space (m²/person) | | | 0.037 |
|   ≤40 (*n* = 1,249) | 17.3 [15.2–19.4] | 82.7 [80.6–84.8] | |
|   >40 (*n* = 106) | 9.4 [3.7–15.1] | 90.6 [84.9–96.3] | |
| How to test the temperature of milk bottle before feeding? | | | < 0.001 |
|   Wrist method (*n* = 1,149) | 15.1 [13–17.2] | 84.9 [82.8–87] | |
|   Mouth testing (*n* = 164) | 27.4 [20.4–34.4] | 72.6 [65.6–79.6] | |
|   Thermometer (*n* = 42) | 16.7 [5.2–28.2] | 83.3 [71.8–94.8] | |
| Infant feeding method | | | 0.002 |
|   Pre-mastication (*n* = 43) | 25.6 [12.3–38.9] | 74.4 [61.1–87.7] | |
|   Sharing cutlery with the feeder (*n* = 158) | 24.7 [17.8–31.6] | 75.3 [68.4–82.2] | |
|   Using child-only cutlery (*n* = 1,150) | 15 [12.9–17.1] | 85 [82.9–87.1] | |
| Fruits were peeled before eating | | | 0.036 |
|   No (*n* = 143) | 10.5 [5.4–15.6] | 89.5 [84.4–94.6] | |
|   Yes (*n* = 1,212) | 17.4 [15.2–19.6] | 82.6 [80.4–84.8] | |
| Sharing toothbrush | | | 0.014 |
|   No (*n* = 1,218) | 15.8 [13.7–17.9] | 84.2 [82.1–86.3] | |
|   Yes (*n* = 137) | 24.1 [16.8–31.4] | 75.9 [68.6–83.2] | |
| Frequency of eating out | | | 0.002 |
|   ≤1 time per week (*n* = 1,074) | 15.8 [13.6–18] | 84.2 [82–86.4] | |
|   2–4 times per week (*n* = 248) | 17.3 [12.5–22.1] | 82.7 [77.9–87.5] | |
|   ≥5 times per week (*n* = 33) | 39.4 [22.4–56.4] | 60.6 [43.6–77.6] | |
| Snacking habit | | | 0.032 |
|   No (*n* = 398) | 13.3 [9.9–16.7] | 86.7 [83.3–90.1] | |
|   Yes (*n* = 957) | 18.1 [15.6–20.6] | 81.9 [79.4–84.4] | |

Note:
Data are presented as % (95% confidence interval). The differences between *H. pylori*-positive and negative children were compared using the chi-square test.

feeder's cutlery, sharing toothbrush, eating out at least 5 times a week, snacking and eating fruits after peeling.

## Protective and risk factors for *H. pylori* infection

Risk factors for *H. pylori* infection were further investigated by univariate and multivariable logistic regression analyses adjusting for age. The final model of multivariable analysis

**Table 3 Univariate and multivariable logistic regression analyses of variables associated with H. pylori infection status.**

| Variables | Univariate | | Multivariable | |
|---|---|---|---|---|
| | OR (95% CI) | P | OR (95% CI) | P |
| **Number of children** | | | | |
| 1 | Reference | | | |
| 2–3 | 1.44 [1–2.07] | 0.05 | | |
| >3 | 2.06 [1.16–3.66] | 0.013 | | |
| **Parents education level** | | | | |
| Both were high school graduates | Reference | | Reference | |
| Either one or both had completed tertiary education | 0.59 [0.43–0.81] | 0.001 | 0.64 [0.46–0.89] | 0.008 |
| **Living space (m²/person)** | | | | |
| ≤40 | Reference | | Reference | |
| >40 | 0.51 [0.26–0.99] | 0.047 | 0.54 [0.28–1.07] | 0.078 |
| **Milk bottle temperature test method** | | | | |
| Wrist method | Reference | | Reference | |
| Mouth testing | 2.05 [1.39–3.01] | <0.001 | 1.79 [1.19–2.68] | 0.005 |
| Thermometer | 1.13 [0.5–2.59] | 0.766 | 1.06 [0.45–2.46] | 0.899 |
| **Infant feeding method** | | | | |
| Using child-only cutlery | Reference | | Reference | |
| Pre-mastication | 1.96 [0.97–3.95] | 0.062 | 1.69 [0.81–3.54] | 0.163 |
| Using feeder's cutlery | 1.86 [1.25–2.77] | 0.002 | 1.84 [1.22–2.78] | 0.004 |
| **Frequency of eating out** | | | | |
| ≤1 time per week | Reference | | Reference | |
| 2–4 times per week | 1.1 [0.76–1.6] | 0.606 | 1.2 [0.82–1.77] | 0.347 |
| ≥5 times per week | 3.23 [1.55–6.73] | 0.002 | 3.13 [1.46–6.68] | 0.003 |
| **Fruits were peeled before eating** | | | | |
| No | Reference | | Reference | |
| Yes | 2.06 [1.14–3.71] | 0.017 | 2.56 [1.4–4.71] | 0.002 |
| **Sharing toothbrush** | | | | |
| No | Reference | | | |
| Yes | 1.6 [1.04–2.46] | 0.032 | | |
| **Snacking habit** | | | | |
| No | Reference | | Reference | |
| Yes | 1.44 [1.03–2.01] | 0.034 | 1.43 [1.01–2.01] | 0.043 |

Note:
OR, odds ratio; CI, confidence interval. Both univariate and multivariable logistic regression analyses were performed to identify risk factors associated with H. pylori infection.

revealed six variables significantly associated with *H. pylori* in children (Table 3). Children whose parent(s) with tertiary education level (OR: 0.64; 95% CI [0.46–0.89]; *P* = 0.008) and larger living space at home (OR: 0.54; 95% CI [0.28–1.07]; *P* = 0.078) were protective factors against *H. pylori* infection. Conversely, testing milk bottle temperature by taking a swig as opposed to dripping a few onto the wrist (OR: 1.79; 95% CI [1.19–2.68];

$P = 0.005$), sharing of cutlery between the feeding person and young children rather than using separate cutlery each during meals (OR: 1.84; 95% CI [1.22–2.78]; $P = 0.004$), and snacking (OR: 1.43; 95% CI [1.01–2.01]; $P = 0.043$) resulted in lower risks for *H. pylori* infection. Only eating fruits peeled than unpeeled (OR: 2.56; 95% CI [1.4–4.71]; $P = 0.002$) and eating out frequently for at least 5 times a week (OR: 3.13; 95% CI [1.46–6.68]; $P = 0.003$) would result in substantially higher risks for *H. pylori* infection.

## DISCUSSION

In the present study, *H. pylori* infection in 1,355 children aged 6–12 years was screened using the non-invasive [13]C-UBT, which has been shown to achieve both sensitivity and specificity of at least 95% in children and adults (*Vaira & Vakil, 2001*). The overall *H. pylori* prevalence was 16.7%, which was at least two-fold higher compared to the global infection rate of 7.9% among children aged 7–12 years from Beijing, Guangzhou and Chengdu (*Ding et al., 2015*). In contrast to Beijing, Guangzhou and Chengdu which are all highly developed cities, Kuichong is only a developing subdistrict of the Shenzhen City. Therefore, the higher *H. pylori* infection rate among Kuichong's children is likely due to the differences in socioeconomic status and living conditions. Our data demonstrated that parent(s) with higher education level and having a larger living space at home are important protective factors against *H. pylori* infection. These findings are consistent with several previous studies from Houston, Czech Republic, Iran and Vietnam that reported children living in a crowded household and whose parent(s) with less education are at significantly greater risks of acquiring *H. pylori* infection (*Malaty et al., 2001*; *Nguyen et al., 2017*; *Nouraie et al., 2009*; *Sykora et al., 2009*).

Poor hygiene practices when feeding the young ones such as giving pre-chewed food and the use of same spoon by both mother and child are among the risk factors of *H. pylori* infection in children (*Nguyen et al., 2017*; *Van Duynhoven & De Jonge, 2001*). Our study demonstrated that testing milk bottle temperature by taking a few sucks directly and using the same cutlery to feed young children would result in a slight but significant increased risk of *H. pylori* infection. This implies that *H. pylori* exists in the human oral cavity and can be transmitted from an infected individual to another person via the oral–oral route (*Yee, 2017*). Hence to reduce the infection risk in children, whoever is the primary carer at home for the young children should exercise good personal hygiene to avoid any food that is to be served to children being cross-contaminated by his/her saliva, especially one who knows him/herself is a *H. pylori* carrier.

In a previous investigation conducted in Peru, children were exposed to greater risk of *H. pylori* infection due to increased consumption of food from street vendors (*Begue et al., 1998*). Similarly, in the present study, frequent dining out was significantly associated with the acquisition of *H. pylori* infection in children, which could be attributed to the preparation of food by food handlers who did not practice good personal hygiene.

While washing and peeling help to remove surface bacteria from fruits and vegetables to reduce the risk of foodborne illness, intriguingly, eating peeled fruits would increase the risk of *H. pylori* infection in this study. As our data showed that consumption of

unboiled water is not associated with *H. pylori* infection, we therefore ruled out our initial thought that the cause of this increased risk might be due to contaminated water source. Unfortunately, we did not question what the fruit was being consumed. Could it possibly be that the fruit peel contains anti-*H. pylori* property and eating the fruit without its skin had therefore resulted in an elevated infection risk? This warrants further investigation. Also, interestingly, children who seemed to snack had a greater chance of acquiring *H. pylori* infection. Again, it was not specified further within the questionnaire on what snacks were being consumed to allow us making a better assumption on the underlying cause.

We concede that there are limitations in our study. Firstly, to assess the risk factors of *H. pylori* infection among children, a cohort study or a case-control study should be conducted, rather than a cross-sectional study which is not the most appropriate option. However, it is important to mention that the basic findings of a cross-sectional study could serve as the foundation for preparing and designing further in-depth case-control studies, cohort studies or randomized controlled trial studies (*Mann, 2003*). Another limitation lies within the questionnaire design as there were questions not explicit and detailed enough to examine the fundamental cause of some risk factors for *H. pylori* infection identified in this study, this warrants improvement before conducting a future cohort study to validate our current findings.

## CONCLUSIONS

There is a high prevalence of *H. pylori* among children aged 6–12 years in Shenzhen City, China. Larger living space at home and parent(s) with tertiary education level were protective factors against *H. pylori* infection in children. Testing milk bottle temperature by taking a swig, not using individual cutlery set when feeding young children and snacking were associated with lower risks for *H. pylori* infection. Only eating fruits peeled than unpeeled and frequent dining out for at least 5 times a week could lead to higher risks for *H. pylori* infection.

### Funding

This work was supported by the Sanming Project of Medicine in Shenzhen (No. SZSM201510050). The funders had no role in study design, data collection and analysis, decision to publish, or preparation of the manuscript.

### Grant Disclosures

The following grant information was disclosed by the authors:
Sanming Project of Medicine in Shenzhen: SZSM201510050.

### Competing Interests

The authors declare that they have no competing interests.

## Author Contributions

- Jingjing Hu performed the experiments, analyzed the data, prepared figures and/or tables, authored or reviewed drafts of the paper, and approved the final draft.
- Xiangyu Wang performed the experiments, analyzed the data, prepared figures and/or tables, authored or reviewed drafts of the paper, and approved the final draft.
- Eng Guan Chua analyzed the data, prepared figures and/or tables, authored or reviewed drafts of the paper, and approved the final draft.
- Yongsheng He conceived and designed the experiments, authored or reviewed drafts of the paper, and approved the final draft.
- Qing Shu performed the experiments, authored or reviewed drafts of the paper, and approved the final draft.
- Li Zeng performed the experiments, authored or reviewed drafts of the paper, and approved the final draft.
- Shiyang Luo performed the experiments, authored or reviewed drafts of the paper, and approved the final draft.
- Barry J. Marshall conceived and designed the experiments, authored or reviewed drafts of the paper, and approved the final draft.
- Aijun Liu conceived and designed the experiments, performed the experiments, authored or reviewed drafts of the paper, and approved the final draft.
- Chin Yen Tay conceived and designed the experiments, analyzed the data, authored or reviewed drafts of the paper, and approved the final draft.

## Human Ethics

The following information was supplied relating to ethical approvals (i.e., approving body and any reference numbers):

The study was approved by the research ethics committee of the First Affiliated Hospital of Shenzhen University, Shenzhen Second People's Hospital.

## Data Availability

The outcomes from the questionnaire are available in Table S1.

## Supplemental Information

Supplemental information for this article can be found online at http://dx.doi.org/10.7717/peerj.8878#supplemental-information.

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
