# Peer review of "Prevalence and risk factors of *Helicobacter pylori* infection among children in Kuichong Subdistrict of Shenzhen City, China"

_PeerJ, doi:10.7717/peerj.8878_

## Round 0.1 · original submission · Major Revisions

As you can see, your submission triggered quite contrasting comments and suggestions, going from minor revision to rejection. Based on my own analysis of your submission and the comments of the reviewers, I wish to give you the opportunity to to come back with a substantially improved version based on the comments of the reviewers. Note that one reviewer questioned the novelty of your work and suggested publication in a more local Journal. While PeerJ does snot make novelty as an essential determinant in final acceptance (thus, the Journal will publish confirmatory results), this does not mean that we will take papers that are of local interest only. So, please, elaborate and document how and to what extent your submission represents a real advance in the topic, and goes beyond descriptions of local interest.

Do not forget to prepare and upload a detailed rebuttal (point by point) in which you indicate clearly how and where you have taken the comments of the reviewers into account in the revised version. If you disagree with some comments and suggestions, explain why. Also, be aware that your revised version may enter a new round of review by the same or by different reviewers. I cannot, therefore, make any commitment at this stage about the final acceptance of your revised version.

·

Basic reporting

I review of the English applied in the manuscript should be revised. Nothing very problematic but there are some sentences not well explicit. Very good manuscript with interesting results. However, I suggest demonstrating the association and non-association with other risk factors, if questioned and analyzed.
The discussion should be more developed and supported in more scientific articles.

Experimental design

Good methodology applied. The questionnaire should be more explicit and the questions should be better described in the Materials and Methods section.

Validity of the findings

Nothing to add.

Additional comments

Congratulations for the present manuscript. The authors have found very interesting results on the risk factors and prevalence of H. pylori among children which is a very important public health issue. I suggest to verify the adding of other risk factors associated with the presence and non-presence of H. pylori and explain the non-association between some risk factors analyzed.

Reviewer 2 ·

Basic reporting

No comment

Experimental design

There is no novelty of this research as the study aims to explore different well known socioeconomic and demographic risk factors for Helicobacter pylori infection and its results only replicate the current knowledge on the subject.
The only part of the study that can be considered as novel is the estimation of the prevalence of Helicobacter pylori infection in a subdistrict of one city in China, therefore the article should be published in a medical journal on a national scale.

Validity of the findings

The validity of the findings for the international audience is questionable .

Reviewer 3 ·

Basic reporting

Your introduction needs more detail to provide more justification for your study.
Remover “prospectivo” word in Objective.

Tables 2 and 3 can be included in a single table.

Introduction, results and discussion. There are a few sentences that need rephrasing for clarity.

Experimental design

Methods
A cross-sectional study is not the most appropriate to assess the risk factors of H. pylori infection among children. The design should be a cohort study or case-control. The methods were not described with sufficient detail.
• Explain how you obtained the sample size.
• How did they do the sampling?
• Prevalence rates should be estimated using 95% CI.

Validity of the findings

Prevalence of H. pylori infection.
Line 129… while 8-year-old children had the highest incidence rate of H. pylori infection at 23.2%... The results must be in terms of prevalence, not incidence.
Prevalence rates should be estimated using 95% CI in each group (general, age and gender)
Tables 1, 2 and 3. Include the statistical test at the bottom of Table.

Risk factors for H. pylori infection. I suggest review interpretations of risk factors for H. pylori infection.
• There si a negative association. Children whose parent(s) with tertiary education level (OR: 0.64; 95% CI: 0.46-0.89; P = 0.008) and larger living space at home (OR: 0.54; 95% CI: 0.28-1.07; P = 0.078) are protective factors for H. pylori infection.
• The OR values are very low to be a high risk for H. pylori infection. Testing milk bottle temperature by taking a swig as opposed to dripping a few onto the wrist (OR: 1.79; 95% CI: 1.19-2.68; P = 0.005), sharing of cutlery between the feeding person and young children rather than using separate cutlery each during meals (OR: 1.84; 95% CI: 1.22-2.78; P = 0.004), and snacking (OR: 1.43; 95% CI: 1.01-2.01; P = 0.043) resulted in lower risks for H. pylori infection.
• Only the eating fruits peeled than unpeeled (OR: 2.56; 95% CI: 1.4-4.71; P = 0.002) and eating out frequently for at least 5 times a week would increase the risk of infection by three times in children (OR: 3.13; 95% CI: 1.46-6.68; P = 0.003) resulted in higher risks for H. pylori infection.

Discussion
Your findings could be expanded with international studies.
The discussion should be adjusted to the corrections of results section (as I have noted above; see results).
Include and discuss the limitations of your study in a paragraph. For example, design study.
Explain you about the shortcomings and limitations C-urea breath test as a diagnostic tool for H. pylori infection.

Conclusión and Abstract
The conclusions section should be appropriately stated and should be limited to those supported by the results. Correct the conclusions according to the comments “Risk factors for H. pylori infection”
Include the prevalence findings. For example, there is a high prevalence of H. pylori among children in Shenzhen City, China.

Additional comments

In this study, the authors to assess the Prevalence and risk factors of Helicobacter pylori infection among children in Kuichong Subdistrict of Shenzhen City, China. Your manuscript is interesting and may contribute to local epidemiology in China. However, this study has several limitations. The main limitation of this study is the design. A cross-sectional study is not the most appropriate to assess the risk factors of H. pylori infection among children. The design should be a cohort study or case-control. To make this paper publishable, the author needs to respond to the comments, as I have noted above.

---

## Round 0.2 · Minor Revisions

Your revised version was considered as acceptable as such by one of the original reviewer but the second one suggested some further very minor modifications. Can you, please, do this and prepare a new revised version ? I look forward hearing from you.

·

Basic reporting

Nothing more do add. The revision was accomplished as suggested by the reviewers.

Experimental design

Nothing more do add. The revision was accomplished as suggested by the reviewers.

Validity of the findings

Nothing more do add.

Additional comments

Nothing more do add. The revision was accomplished as suggested by the reviewers. I believe that the paper is ready for final publication process.

Reviewer 3 ·

Basic reporting

No comments.

Experimental design

No comments.

Validity of the findings

Abstract and Results sections. The overall H. pilory prevalence rate (16.7%) should be estimated using 95% CI.

Conclusion. The sentence “In the Kuichong Subdistrict of Shenzhen City, China, 16.7% of children aged 6-12 years were found to be infected with H. pylori” is repetitive. Consider “there is a high prevalence of H. pylori among children aged 6-12 years in Shenzhen City, China”.

Additional comments

No comments.

---

## Round 0.3 · accepted · Accept

Thank you for making the additional last corrections. I believe this will make your paper even better.